# Health Risk and Underweight

**DOI:** 10.3390/nu15143262

**Published:** 2023-07-24

**Authors:** Ulrich Cuntz, Norbert Quadflieg, Ulrich Voderholzer

**Affiliations:** 1Center for Psychosomatic Medicine, Schoen Klinik Roseneck, Am Roseneck 6, 83209 Prien am Chiemsee, Germany; uvoderholzer@schoen-klinik.de; 2Forschungsprogramm für Psychotherapieevaluation im Komplexen Therapiesetting, Paracelsus Medizinische Universität, 5020 Salzburg, Austria; 3Departamento de Enlace y Medicina Psicosomatica, Universidad Catolica, Santiago 8831314, Chile; 4Department of Psychiatry and Psychotherapy, Ludwig Maximilians Universität, 80336 Munich, Germany; norbert.quadflieg@med.uni-muenchen.de

**Keywords:** anorexia nervosa, eating disorders, underweight, health risk, morbidity, mortality percentiles

## Abstract

Anorexia nervosa is associated with a significant risk of morbidity and mortality. In clinical practice, health risk is assessed and estimated using routinely collected laboratory data. This study will develop a risk score using clinically relevant laboratory parameters. The related question is how to estimate the health risk associated with underweight using body weight, height and age. Methods: We used routinely collected laboratory parameters from a total of 4087 patients. The risk score was calculated on the basis of electrolytes, blood count, transaminases and LDH. The nine parameters used were summed as zlog-transformed values. Where appropriate, the scales were inverted so that high values represented higher risk. For statistical prediction of the risk score, weight/height and age reference values from the WHO, the CDC (Center of Disease Control) and representative studies of German children and adults (KIGGS and NNS) were used. Results: The score calculated from nine laboratory parameters already shows a convincing relationship with BMI. Among the weight measures used for height and age, the z-score from the CDC reference population emerged as the best estimate, explaining 34% of the variance in health risk measured by the laboratory score. The percentile rank for each age-specific median weight from the KIGGS/NNS still explained more than 31% of the variance. In contrast, percentiles explained less variance than BMI without age correction. Conclusions: The score we used from routine laboratory parameters appears to be an appropriate measure for assessing the health risk associated with underweight, as measured by the quality of the association with BMI. For estimating health risk based on weight, height and age alone, z-scores and percentages of age-specific median weight, as opposed to percentiles, are appropriate parameters. However, the study also shows that existing age-specific BMI reference values do not represent risk optimally. Improved statistical estimation methods would be desirable.

## 1. Introduction

Both being overweight and being underweight is associated with an increased health risk. However, the increase in health risk is much greater with increasing underweight than with increasing overweight. While overweight and obesity pose a significant health risk, extreme underweight and anorexia nervosa (AN) are associated with an even higher health risk.

For the question of the risk associated with obesity, the large epidemiological population-based studies that have assessed the risk of mortality are well established [1,2,3]. The body mass index (BMI) has been shown to be a useful criterion for assessing risk in these populations. However, the prevalence of obesity is much higher than the prevalence of underweight, especially in its extreme forms. Therefore, mortality is not an appropriate risk indicator for extreme underweight and anorexia nervosa. Risk indicators generated based on laboratory values are probably better suited for this purpose.

In daily clinical practice, we use a number of laboratory parameters to assess individual health risk in AN. Typical changes in electrolytes, hematological parameters, and enzymes worsen with the degree of underweight, and indicate a significantly increased health risk. Although the laboratory parameters that are usually routinely collected do not capture the full health risk associated with underweight, they nevertheless appear to be a comparatively simple and suitable method for everyday clinical practice to assess the health risk for the individual patient.

The healthy weight range for a given body height varies considerably with age. This is particularly relevant for children and adolescents but is also likely to be relevant for adults. Based on epidemiological data, normative values have been calculated for different populations, which allow the BMI available for different age groups to be converted into Z-values and percentiles based on the respective deviation from the median value. It is also common to convert the available BMI into a percentage of the expected median weight for the age group. Whether these formulae are suitable for adequately predicting the health risk in AN has not yet been sufficiently clarified.

### 1.1. Risk Indicators for Underweight and Anorexia Nervosa

Electrolytes and water balance: Both dehydration and overhydration can occur in an eating disorder. Both can put the person at great risk and need to be properly regulated and normalized appropriately. It is also important to know that significant changes in body weight can occur without changes in body substance such as when the amount of water present in the body changes.

Several factors lead to a water imbalance in eating disorders. Recurrent vomiting, the use of diuretics or the frequent use of laxatives lead to a loss of sodium as well as water. Water loss usually affects not only the extracellular space, but also the intracellular volume. In addition to insufficient food intake, the regulation of fluid intake is also very often disturbed: some patients suppress or ignore their feeling of thirst and drink far too little; others drink a lot to suppress the feeling of hunger. The active transport mechanisms of the cell wall suffer from the latent or manifest energy deficit, leading to a decrease in cell turgor and electrolyte gradients. Dehydration due to purging and inadequate fluid intake therefore affects not only the extracellular space but also the cell volume.

When the body is in a state of prolonged stress (e.g., due to lack of energy), the HPA axis is overactivated [4], which often also leads to an increased release of vasopressin (antidiuretic hormone) [5]. This is exacerbated by the electrolyte loss caused by purging. This over-activation of the HPA axis and increased aldosterone secretion can lead to pseudo-Bartter syndrome, a constellation of signs/symptoms that include oedema, metabolic alkalosis and low serum potassium.

**Hypokalemia:** Loss of potassium due to vomiting, diuretic or laxative abuse or increased potassium excretion in the context of hyperaldosteronism initially affects the extracellular space, while the intracellular potassium concentration is less affected. This leads to an increase in the potassium voltage of muscle and nerve cells and thus to an increased electrophysiological excitability of these cells. Hypokalemia associated with AN or diuretic abuse has been blamed in case studies for the severe complication of pontine myelinolysis [6], although in these cases there was no concomitant hyponatremia. Similarly, there are case reports of rhabdomyolysis [7]. Chronic hypokalemia associated with volume deficiency and substrate depletion may contribute to the development of hypokalemic nephropathy [8]

**Hyponatremia** is a relative excess of water in relation to sodium. It can be caused by a marked increase in water intake (primary polydipsia) and/or by impaired water excretion, due, for example, to advanced renal insufficiency or prolonged secretion of antidiuretic hormone (ADH).

The more acute the hyponatremia, the greater the risk of complications. Serum sodium levels above 130 mmol/L (or 130 mEg/L) are considered “mild hyponatremia”; levels below 120 mmol/L (or 120 mEq/L) are considered severe. Clinical symptoms include nausea, malaise, headache, lethargy and possibly seizures. In extreme cases of hyponatremia, coma and respiratory arrest (<115 to 120 mEq/L) and noncardiogenic pulmonary oedema may occur. Extreme hyponatremia can lead to herniation of the brainstem, which is the most severe complication of hyponatremia [9,10].

**Hypophosphatemia** has adverse effects on all organs and systems of the body [11]. In the heart muscle, impaired contractility and reduced cardiac output can lead to ventricular arrhythmias and heart failure [12,13]. Muscle dysfunction in a variety of tissues can lead to ophthalmoplegia, dysphagia or ileus; rhabdomyolysis can cause severe muscle pain and weakness or acute renal tubular necrosis. Impaired neurological function may include confusion, delirium, seizures, tetany or coma [12,14].

### 1.2. Bone Marrow Suppression

Eating disorders are associated with characteristic pathological changes in routine laboratory values. In patients with AN, the bone marrow is often hypoplastic and gelatinous [15,16], which can lead to both leukopenia and anemia [17,18]. Both anemia and leukopenia normalize with weight gain and usually do not require separate treatment.

Leukopenia is more pronounced in higher degrees of underweight and mainly characterized by a decrease in the number of granulocytes. It can be assumed that people with severe underweight are at increased risk of bacterial infections due to leukopenia. Another characteristic feature is anemia, which is aplastic in almost all cases. Although oral iron intake is often reduced in AN, iron losses are lower due to amenorrhea, so iron deficiency is less likely to be the cause of anemia.

This is also associated with a drop in hematocrit, often additionally caused by an increase in extracellular volume. The causes of this are described above. In these cases, hyponatremia and a decrease in hematocrit occur in parallel.

### 1.3. Autolysis

Elevated transaminases are a common laboratory finding in patients with AN and are associated with the degree of malnutrition [19,20,21]. Life-threatening liver failure can occur in extremely underweight patients with AN [22,23,24]. In such cases, coagulopathy (INR > 1.5) and hepatic encephalopathy are added to the markedly elevated liver values. This is initially mild and manifests itself in behavioral changes, mild confusion, slurred speech, and sleep disturbance, but can progress to coma. Nutritional rehabilitation leads to rapid improvement in liver function. As transaminases normalize, INR and encephalopathy improve. (A decrease in transaminases could indicate progression of liver failure but would be associated with an increase in INR and worsening of encephalopathy).

The mechanism behind abnormal liver function tests appears to be hepatic autophagy. The cause of autophagy and the resulting elevated transaminases is the severe energy deprivation associated with AN, which forces the liver cells to obtain energy by self-digestion. Histology typically shows no portal fibrosis or periportal inflammatory infiltrates, and sonography shows a normal echogenicity pattern with no abnormalities. In contrast, microscopy shows autophagosomes in vesicular intracellular compartments containing sequestered material from cell organelles, such as mitochondria or the endoplasmic reticulum. Several authors have shown that the degree of malnutrition correlates with the level of transaminases and, most importantly, that increased food intake leads to a rapid improvement in elevated transaminases. Similarly, lactate dehydrogenase, which is present in all body cells, indicates the death of cells throughout the body due to energy deprivation [25,26,27].

In a previous study on this topic, we summed up the raw pathological laboratory parameters [27]. With this method, the degree of pathological deviation is not taken into account, and thus, essential information is lost. Other approaches, like summing up the z-transformed values or factor analysis, do not consider the considerable skewness of the distribution of pathologically changed laboratory parameters or do not provide a uniform general factor due to the low communalities of the parameters.

One aim of the study was to present a health risk score for AN derived from laboratory parameters that overcomes the above limitations. Another aim was to statistically predict the newly computed health risk score from several different weight-for-age and height measures.

## 2. Materials and Methods

### 2.1. Subjects

This study included the admission laboratory examinations of 4474 patients (aged between 12 and 73 years) with an admission diagnosis of AN, who were treated in an inpatient unit at the Schön Klinik Roseneck in Prien am Chiemsee between August 2014 and March 2022. Some patients were already weight normalized on admission and were admitted for maintenance treatment (Table 1).

Almost all patients in the sample were significantly underweight. Clinically, underweight is a sign of an underlying organic disease. On the other hand, the diagnosis of AN implies a psychological genesis of the underweight. As a rule, patients are referred to the clinic when other causes of underweight have been ruled out, so that we can assume that the observed underweight has no organic causes.

The intake values for GOT, GPT, LDH, leucocytes, hematocrit, hemoglobin, sodium, potassium, and phosphate were used as risk indicators. A total of 4087 patients had all these parameters at admission (within the first 5 days after admission) and were included in the study.

### 2.2. Zlog Transformation

We have zlog-transformed the laboratory parameters in order to be able to sum up the laboratory values and calculate a risk score. This gives a relative value that indicates by how many standard deviations a measured value differs from the mean value of the reference population [28].

The zlog value is easy to interpret: its reference interval is between −1.96 and +1.96, independently of the method; strongly reduced or increased laboratory results lead to zlog values around −5 or +5. As the zlog transformation leads to the same deviation measures for all laboratory values, it allows for the summation of the individual laboratory results that is necessary for our purposes.

For GOT, GPT and LDH, higher values predict an increased risk of morbidity and mortality. In contrast, the electrolyte levels of potassium, sodium and inorganic phosphate and the blood levels of hematocrit, hemoglobin and leukocytes are typically lower in more extreme underweight and are associated with a higher risk. For this reason, we have inverted the zlog values of the electrolytes and the blood values (multiplied by −1), so that a higher value always indicates an increased risk. The zlog values calculated in this way are then simply added to give a health risk score.

### 2.3. Weight for Height and Age

The health risk associated with BMI, or the BMI range with the lowest risk, varies considerably with age. Population data are available for many countries around the world. For the purpose of this study, the BMI for Height tables from the WHO [29,30], the CDC [31,32], the German KIGGS study [33,34] and the second German National Nutrition Survey (NNS; also gives percentiles for adults) [35] were used. 

The 2000 CDC (Center for Disease Control and Prevention) Growth Curves are based on health data from the USA and have been providing age-appropriate BMI curves since 2000. They were designed primarily to detect children at risk of becoming overweight at an early age. The WHO Multicentre Growth Reference Study (MGRS) was conducted between 1997 and 2003 to establish new growth curves to assess the growth and development of infants and young children worldwide. The MGRS collected primary growth data and related information from about 8500 children from very different ethnic backgrounds and cultures (Brazil, Ghana, India, Norway, Oman and the USA). Reference values from Germany are provided by the KiGGS baseline survey, which was carried out between 2003 and 2006 in a total of 167 towns and municipalities in Germany with a clustered random sample of 17,641 children and adolescents aged 0 to 17 years [36].

The data basis for the calculation of the BMI reference values for adults is the Second National Nutrition Survey Germany (NNS II), which was collected in 2005 and 2006. This study is representative for the German population and is based on more than 13,000 adults [35,37]. We combined these reference values with the reference values of the KIGGS study (KIGGS/NNS) and thus offer reference values for the entire age spectrum of our patients. Since most of the patients with AN are below the 3rd percentile and thus no values are provided in the usual charts, z-score and percentile were calculated according to the LMS method (L = skewness, M = median, S = deviation). The distribution of the BMI in the age groups is considerably left-skewed, so that BMI values deviating far from the median are much rarer for low body weight than for weight deviating upwards.

With the aid of the LMS method, Z-values can be calculated by considering not only the median (M) and the standard deviation (S) but also the skewness of the distribution (L). By including the parameter L in the calculation (z = (measure/M)L − 1/(L/S)), the considerably left-skewed distribution of BMI in the age groups can be corrected [38].

The percentiles can be calculated directly from the z-scores.

Accordingly, the following parameters were calculated for the following analyses:Z-value;Percentile;Percentage of the current weight in relation to median weight for the respective age.

These parameters were calculated from the LMS values of the CDC, the WHO and the data of the KIGGS and the NNS. KIGGS and NNS data also allow for the calculation of age-specific Z-values and percentiles for the adult patient groups. For the CDC and WHO, the highest age available in each case were used for all adults (240 months = 20 years).

For the variance explanation of the risk score we calculated, we had 9 different weight for age for height formulas available in addition to the BMI: z-score, percentile and percent of median weight for CDC, WHO and KIGGS/NNS.

## 3. Results

Based on clinical knowledge, we selected nine laboratory parameters that are particularly indicative of the health risks associated with being underweight. The individual laboratory parameters correlate only insignificantly or moderately with the BMI. On the other hand, the sum score calculated using the zlog transformation shows a very close and strong significant correlation with BMI. With a Pearson correlation of 0.427 (*p* < 0.001; N = 4087), the admission BMI already correlates very highly with the risk score calculated from the nine laboratory parameters mentioned. This sets a benchmark for further analysis of the BMI for age parameters available to us. The consideration of age through population-based norms must be better than that by BMI alone. The following graph (Figure 1) shows the close correlation between BMI and health risk measured by laboratory values.

The z-scores and percentiles calculated according to the LMS method initially produce results that are quite obviously implausible. A z-score of −6 gives a percentile rank of 0.000000099. Accordingly, in a country with more than 80 million inhabitants, like Germany, one would expect to find about 50 people with such a low BMI. It would be unlikely to find any people with a z-score of −7. In our sample, more than 20% ( *n* = 978) of patients had a z-score of −6 and below. The lowest z-score was a statistically completely implausible value of −27. For example, a female 32-year-old patient with a BMI of 8 had a z-score of −25 according to the CDC tables. The ten logarithms of this z-score give a percentile rank of 10^−135^. Despite this apparently implausible classification of our patient group, the possibility of an age correction of the measured BMI arises from these population-based parameters.

Regression analysis with the BMI reference values available to us shows that more than 30 percent of the variance of the health risk identified by the laboratory values can be explained by the age-adjusted BMI (see Table 2). The highest correlation is provided by the z-score based on the CDC data. 

However, the variance explained by the KIGGS data is comparably high. In this case, we also used the NNS tables to convert the BMI of the adult patients into z-values, percentiles and % median values. The age of the adult patients was also taken into account (instead of using the LMS values for 240 months as was used by the WHO and CDC).

It is obvious that percentiles derived from one of the samples used do not correlate very well with risk and provide little explanation for the variance.

The percentage value of the respective age-specific median weight has a somewhat lower correlation and thus variance explanation of the health risk. However, the correlation is much higher than that of the percentiles.

Percentiles, z-values and %-median weight are used to adjust BMI according to age. If we perform a regression analysis with the z-value of the CDC and the respective age, the percentage of explained variance increases again significantly; even the best parameter we used in terms of variance clarification, the z-value CDC, ‘only’ has a variance clarification of slightly more than 30% (R^2^ = 0.341). If a regression analysis for the measurable risk is carried out with this z-value and additionally with age, the explanation of variance increases to more than 40% (r = 0.652, R^2^ = 0.425).

The data provided by Hemmelmann [35] allow for z-transformation, the calculation of percentiles and also the indication of %Median values for the large group of adult patients with anorexia nervosa. If one compares the variance explanation by the BMI alone with the population-based parameters, one sees that the variance explanation can also be significantly improved (Table 3). The risk score is significantly higher in the elderly than in younger patients with comparable BMI. This means that underweight in the elderly is associated with higher health risk. A mean value of the risk score in patients under 20 years of age and with a BMI under 13 is 7.3, which is lower than a risk score of 15.1 in patients over 40 years of age with the same weight.

The following graph (Figure 2) shows the correlation between the health risk that can be assessed based on the laboratory values and the CDC z-score.

## 4. Discussion

The zlog transformation method presented here provides a pragmatic approach to solving a difficult problem: for a score that calculates the indexed risk from a set of laboratory parameters, the problem arises of how to calculate a sum score from the individual laboratory parameters. As all the laboratory parameters have the same median and the same dispersion parameters with the zlog transformation, a summation is possible. In addition, changes in laboratory methodology leading to altered reference values can be taken into account so that the zlog-transformed parameters are directly comparable. The risk score calculated using the zlog transformation is based on health risk indicators used in clinical practice for severely underweight female patients with anorexia nervosa. In the form used, these are only indicators and not a measure of actual morbidity, complications that occur, or even mortality. However, the high correlation between the score and the indicated risk shows that the measurable consequences of being underweight can indeed be captured. This score opens the possibility of investigating the extent to which the reference values for weight for body weight and age obtained from the respective populations are suitable for estimating the extent of underweight in terms of the associated health risk.

Although the fact of increased morbidity and mortality in anorexia nervosa has long been known [39,40,41], to our knowledge, there have been few attempts to capture this risk in a risk score. For the most severe underweight (below a BMI of 13), we had used a risk score that added up the number of pathological values [27]. The methodology used here, which allows the full range of variance in laboratory values to be considered, is clearly superior to the risk score used in our previous study.

The validity of z-scores and percentiles has been questioned elsewhere [27,42]. This is particularly true for extreme percentile ranks above the 95th or below the 5th percentile. This is probably due to the fact that the population samples are adjusted for those cases that seem biologically implausible. However, these are obviously the very cases that could provide realistic percentages in the case of AN.

Correcting body weight for height and age is particularly useful if the aim is to obtain an estimate of the health risks associated with body weight. Most of the work in this area has been conducted to estimate the risks of overweight and obesity. Although the risk of being underweight seems to be much higher than that of being overweight, the risk of being overweight has been studied much more intensively. This is probably due to overweight being so much more common than underweight.

Although the calculated z-scores and percentiles obviously lead to unrealistic results, they do allow for the explanation of a considerable part of the variance of the health risk related to underweight that can be measured by laboratory values. Z-values calculated according to CDC or KIGGS-NNS explain more than 30% of the variance. Z-Scores based on WHO data are significantly worse. Percentiles do not adequately reflect the health risk in AN. The variance explanation is lower than for the non-age-corrected BMI.

Using the parameter-free data as percentages of the age-specific median weight, the variance explanation with the KIGGS-NNS medians is almost as good as with the Z values. The median weights from the WHO and CDC samples perform worse.

The National Nutrition Survey also provides reference values for adults in Germany, which allow for the age-specific weighting of body weight. Even if the advantage over BMI is not nearly as great as for adolescents, the improved variance explanation shows that health risks is age-dependent even within the adult part of the sample. This applies both to the z-scores and to the percentages of median weight.

Although a variance explanation of more than 30% of the measurable health risk seems high, the proportion of the health risk accounted for by the degree of underweight may be even significantly higher. An important question is whether the BMI formula which divides the body weight by the square of the body height in meters, is the best formula for adjusting the weight to the respective body height. If weight is divided by height to the power of 2.4 instead of 2.0, the correlation with risk increases from 0.427 to 0.432.

However, the way in which the degree of underweight is assessed for the respective age is probably much more important. As none of the reference samples used from KIGGS/NNS, WHO or CDC are representative of the AN population, the distribution measures obtained from these samples are not suitable for correctly estimating the degree of underweight. There are three reasons for this:The patient weight as a percentage of the age-specific median weight, which does not use control measures, explains almost as much variance as the z-score.The percentiles calculated from the z-scores are completely unrealistic.In a regression analysis on the risk score based on the z-score and including age as a predictor, the explained variance increases significantly to over 40%. This means that the z-score does not provide all the necessary information about the age specificity of the respective BMI.

Obviously, there is also an age specificity of the health risk associated with the degree of underweight in adults.

## 5. Conclusions

Population-based reference samples do not include a sufficient number of very underweight people to provide realistic measures of prevalence. This applies to the KIGGS/NNS, WHO and CDC samples used here. However, the problem is likely to be the same in other reference samples. As the health risk associated with being underweight is very high (the diagnosis is associated with a higher mortality risk than in the normal population [39,43]), the assessment of the health risk associated with the degree of underweight is very important for the management of this group of people.

Our study shows that the best parameter for assessing health risk is the z-score (according to the distribution parameters of the CDC reference values). However, the percentage value of the expected median weight provides similar information and is easier to calculate in clinical practice.

The study also points to a fundamental problem in the grading of underweight and the associated health risk. The population-based reference values are unlikely to provide sufficient information for this small part of the population in the future. The statistical basis for grading underweight needs to be reconsidered.

## Figures and Tables

**Figure 1 nutrients-15-03262-f001:**
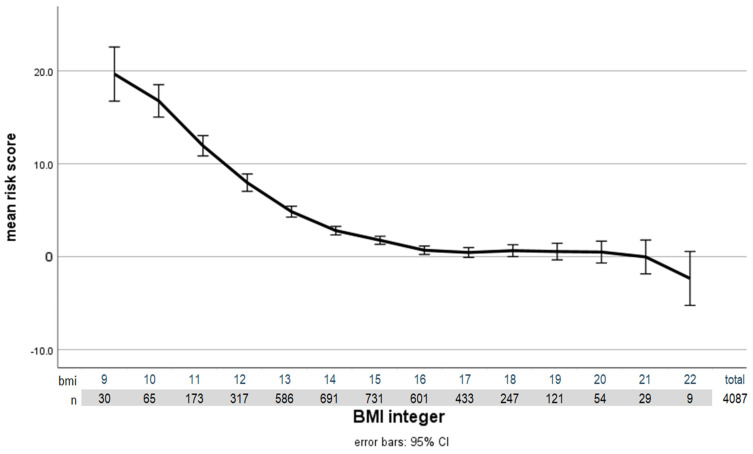
Correlation of health risk (based on laboratory values) with BMI at admission (independent of the respective age). Error bars represent the 95% confidence interval. Patients with a BMI below 9 were subsumed into those with a BMI of 9.

**Figure 2 nutrients-15-03262-f002:**
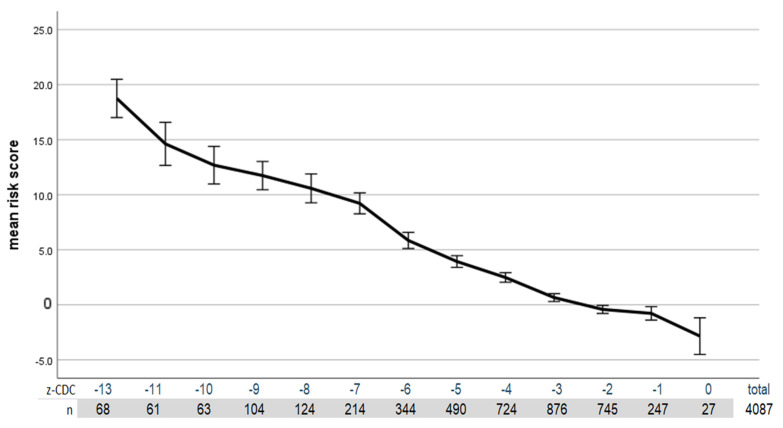
Correlation of health risk (based on laboratory values) with the z-score CDC. The error bars represent the 95% confidence interval. Patients with a z-score below −13 were combined, and those with a z-score of −11 and −12 were combined to achieve sufficient cell occupancy.

**Table 1 nutrients-15-03262-t001:** Age and sex of all inpatients admitted to Schoen Klinik Roseneck between 2014 and 2022.

	All Patients with AN *n* = 4474	With Complete Laboratory *n* = 4087
Female	4288 (95.8%)	3915 (95.8%)
Male	186 (4.2%)	172 (4.2%)
Mean age	22.68	22.76
SD	9.84	9.96

**Table 2 nutrients-15-03262-t002:** Regression analyses for the risk score by the weight for height for age parameter used.

Reference Sample	BMI for Age	r	R^2^	ANOVA F/df	*p*
BMI admission		0.427	0.182	909.14/1	<0.001
KIGGS/NNS	z-value	0.564	0.318	1905.20/1	<0.001
	percentile	0.204	0.042	167.97/1	<0.001
	% median	0.559	0.313	1861.54/1	<0.001
WHO	z-value	0.512	0.262	1447.56/1	<0.001
	percentile	0.218	0.048	204.40/1	<0.001
	% median	0.504	0.254	1389.14 /1	<0.001
CDC	z-value	0.584	0.341	2113.84/1	<0.001
	percentile	0.233	0.054	233.60/1	<0.001
	% median	0.538	0.290	1666.11/1	<0.001

Regression analysis of zlog-transformed risk score by admission BMI and z score, percentile and percentage of median weight derived from the German KIGGS/NNS study, the WHO and CDC. The total sample of inpatients included 4087 valid cases. r: Correlation coefficient, R^2^: variance explained, ANOVA F: F-value variance analysis, *p*: significance variance analysis.

**Table 3 nutrients-15-03262-t003:** Regression analyses for the risk score by BMI at admission and according the BMI-Charts for adults out of the Second National Nutrition Survey—Germany [35]. The total sample of adult (≥18 y) inpatients included 2515 valid cases.

Reference Sample	BMI for Age	r	R^2^	ANOVA F/df	*p*
BMI admission		0.534	0.285	1003.08/1	<0.001
NNS	z-value	0.576	0.332	1250.65/1	<0.001
	percentile	0.232	0.054	143.41/1	<0.001
	% median	0.568	0.322	1194.91/1	<0.001

r: Correlation coefficient, R^2^: variance explained, ANOVA F: F-value variance analysis, *p*: significance variance analysis.

## Data Availability

In accordance with the data security guidelines of our facility, the Schön Klinik Roseneck, data relating to patients must not be publicly accessible.

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
