# Peer review of "Health Risk and Underweight"

_nutrients, 2023, doi:10.3390/nu15143262_

Round 1

Reviewer 1 Report

In this manuscript, Cuntz et al. present the study on Health risk and underweight. The authors examined examines how to develop a measure of health risk associated with the degree of underweight using relevant laboratory parameters and estimate the health risk associated with underweight using body weight, height, and age. Using routinely collected laboratory parameters from a total of 4087 patients, a risk score for anorexia nervosa was calculated based on electrolytes, blood counts, transaminases, and LDH. Finally, they found that the score used from routine laboratory parameters appears to be an appropriate measure for assessing the health risk associated with underweight, as measured by the quality of the association with BMI. This study is interesting, and the experiments are well thought out and executed. The authors should improve the writing skills before its publication.

In this manuscript, Cuntz et al. present the study on Health risk and underweight. The authors examined examines how to develop a measure of health risk associated with the degree of underweight using relevant laboratory parameters and estimate the health risk associated with underweight using body weight, height, and age. Using routinely collected laboratory parameters from a total of 4087 patients, a risk score for anorexia nervosa was calculated based on electrolytes, blood counts, transaminases, and LDH. Finally, they found that the score used from routine laboratory parameters appears to be an appropriate measure for assessing the health risk associated with underweight, as measured by the quality of the association with BMI. This study is interesting, and the experiments are well thought out and executed. The authors should improve the writing skills before its publication.

Author Response

Thank you for your friendly comments – In accordance with your advice, we have revised the language of the entire manuscript.

Reviewer 2 Report

In this paper, the authors used relevant laboratory parameters to develop a measure of health risk associated with the degree of being underweight and estimated the health risk associated with being underweight using body weight, height, and age.  

The paper is interesting and convincing. The analysis of the score obtained from the laboratory data, as well as the weight for height and age in the estimation of health risk, are very useful for the management of underweight patients, such as patients with anorexia nervosa. Moreover, the study revealed that existing age-specific BMI reference values do not optimally represent health risks in extremely underweight patients.

Minor comments

-          Hypokalemia and its abbreviation HK are not indicated in the text as subchapters (as Hyponatremia, Hypophosphatemia).

-          z-log is written in two ways: z-log and zlog.

-          in the discussion chapter, no other studies are presented that used similar risk estimates, and the advantages of the proposed score, compared to other methods, are not highlighted.

Author Response

Thank you for your valuable comments. We would like to answer as follows: 

Minor comments

-          Hypokalemia and its abbreviation HK are not indicated in the text as subchapters (as Hyponatremia, Hypophosphatemia).

Please excuse the misleading abbreviation: HK stands for hematocrit and we have now abandoned this abbreviation. There is now a paragraph dedicated to hypokalemia, analogous to hyponatremia ad hypophosphatemia..

-          z-log is written in two ways: z-log and zlog.

 Thanks for the tip: we have now consistently used the spelling 'zlog'.

-          in the discussion chapter, no other studies are presented that used similar risk estimates, and the advantages of the proposed score, compared to other methods, are not highlighted.

To our knowledge, there have been no attempts by other scientists to develop a risk score. We have therefore referred to our own risk score, which has been decisively further developed with the zlog transformation:

“Although the fact of increased morbidity and mortality in anorexia nervosa has long been known39-41, to our knowledge there have been few attempts to capture this risk in a risk score. For the most severe underweight (below a BMI of 13), we had used a risk score that added up the number of pathological values42. The methodology used here, which allows the full range of variance in laboratory values to be taken into account, is clearly superior to the risk score used there.”

Reviewer 3 Report

Dear authors,

Regarding the submitted manuscript entitled "Health risk and underweight". 

The topic is very interesting and the following revisions should be considered:

1. Page 5, line 216: The formula should be written in words instead of a screenshot/photo.

2. Figure 1 and Figure 2 have blurred "X" axis and not clear enough for the reader.

3. Which disorders/disease/syndromes were the underlying etiologies for anorexic patients? The selection bias could exist while analyzing the patients with different severity of Anorexia Nervosa. I suggest to address your justification in the discussion section.

The written English is fine.

Author Response

Thank you for your valuable comments: 

We have revised the language of the manuscript once again. 

  1. Page 5, line 216: The formula should be written in words instead of a screenshot/photo.

We have changed the paragraph as follows:

“BMI is significantly skewed across age groups. With the aid of the LMS method, Z-values can be can be calculated by considering not only the median (M) and the standard deviation (S) but also the skewness of the distribution (L).  By including the parameter L in the calculation (z=(measure/M)L-1/(L/S)), the considerably left-skewed distribution of BMI in the age groups can be corrected38.

The percentiles can be calculated directly from the z-scores.”

  1. Figure 1 and Figure 2 have blurred "X" axis and not clear enough for the reader.

We tried to improve the readability of the x-axis (see manuscript).

  1. Which disorders/disease/syndromes were the underlying etiologies for anorexic patients? The selection bias could exist while analyzing the patients with different severity of Anorexia Nervosa. I suggest to address your justification in the discussion section.

 We have amended the following paragraph. We hope this addresses your concerns:

“Almost all patients in the sample are significantly underweight. Clinically, underweight is a sign of an underlying organic disease. On the other hand, the diagnosis of anorexia nervosa implies a psychological genesis of the underweight. As a rule, patients are referred to the clinic when other causes of underweight have been ruled out, so that we can assume that the observed underweight has no organic causes.”
